# A molecularly defined subpopulation of oligodendrocyte precursor cells controls the generation of myelinating oligodendrocytes during postnatal development

**Shayan Moghimyfiroozabad[1], Maela A. Paul[1], Lea Bellenger[2], Fekrije Selimi[1] ***

**1** Center for Interdisciplinary Research in Biology (CIRB), College de France, CNRS, INSERM, Université PSL, Paris, France, **2** ARTbio Bioinformatics Analysis Facility, Sorbonne Université, Inserm U1156, CNRS FR 3631, Institut Français de Bioinformatique (IFB), Paris, France

* fekrije.selimi@college-de-france.fr

**Data Availability Statement:** All relevant data are within the paper and its Supporting Information

## Abstract

Oligodendrocyte precursor cells (OPCs) are a class of glial cells that uniformly tiles the entire central nervous system (CNS). They play several key functions across the brain including the generation of oligodendrocytes and the control of myelination. Whether the functional diversity of OPCs is the result of genetically defined subpopulations or of their regulation by external factors has not been definitely established. We discovered that a subpopulation of OPCs found across the brain is defined by the expression of *C1ql1*, a gene previously described for its synaptic function in neurons. This subpopulation starts to appear during the first postnatal week in the mouse cortex. Ablation of *C1ql1*-expressing OPCs in the mouse leads to a massive lack of oligodendrocytes and myelination in many brain regions. This deficit cannot be rescued, even though some OPCs escape *Sox10*-driven ablation and end up partially compensating the OPC loss in the adult. Therefore, *C1ql1* is a molecular marker of a functionally non-redundant subpopulation of OPCs, which controls the generation of myelinating oligodendrocytes.

## Introduction

Oligodendrocyte precursor cells (OPCs) are a type of glial cells that homogeneously populate the central nervous system (CNS) starting during postnatal development and throughout adulthood [1]. Several functions have been attributed to OPCs [2]. They have been documented to control vascularization and angiogenesis [3], engulfment of axons and refinement of neural circuits [4,5], and antigen presenting [6]. However, a major, and their initially described, role is to differentiate into oligodendrocytes (OLs) [7–9]. In rodents, the generation of OLs from OPCs starts at birth and continues during adulthood [10,11]. OPC differentiation peaks during the second postnatal week and then decreases gradually [10]. A portion of OPCs continues to proliferate to maintain the density of their population constant throughout the mature brain [1]. OPCs are thus essential for proper development and function of brain circuits.

files. Custom codes used in this study are found at: DOI 10.5281/zenodo.11065190 and DOI 10.5281/zenodo.11065401.

**Funding:** This work was supported by funding from: European Research Council ERC consolidator grant SynID 724601 (to FS; https://erc.europa.eu/homepage), Agence Nationale de la Recherche (ANR) MEMO LIFE grant ANR-10-LABX-54 (to FS; https://www.memolife.bio.ens.psl.eu/), Institut National du Cancer (INCA) grant PEDIAHR21-014 (to FS; https://en.e-cancer.fr/). SM received a PhD thesis funding from la Ligue Nationale Contre le Cancer (https://www.ligue-cancer.net/) and Ecole des neurosciences de Paris Ile-de-France (ENP). Since 2019, ENP has changed to la Fondation des Neurosciences Paris (FNP, https://www.sorbonne-universite.fr/universite/fondation-sorbonne-universite/fondation-abritee-fondation-des-neurosciences-de-paris). The funders did not play any role in the study design, data collection and analysis, decision to publish, or preparation of the manuscript.

**Competing interests:** The authors have declared that no competing interests exist.

**Abbreviations:** BAI3, brain angiogenesis inhibitor 3; CNS, central nervous system; COP, committed oligodendrocyte precursor; NFOL, newly formed oligodendrocyte; GO, Gene Ontology; MBP, myelin basic protein; MS, multiple sclerosis; OPC, oligodendrocyte precursor cell; PBS, phosphate-buffered saline; PCA, principal component analysis; PDGFRa, platelet-derived growth factor receptor α; smFISH, single-molecule fluorescence in situ hybridization; SNN, shared K-nearest neighbor; UMAP, uniform manifold approximation and projection.

OPCs are generated in the CNS through sequential waves and in a ventro-dorsal manner [7]. In the forebrain, 3 waves with different spatial origins and genetic lineages generate OPCs [12]. The first 2 waves are originally ventral and embryonic. The third wave is generated dorsally after birth [12]. The elimination of each wave of OPCs is rapidly compensated by the other waves without any significant consequences on OPC dynamics or myelination of the developing brain [12]. Even the ablation of all 3 waves in the forebrain is compensated by OPCs coming from the diencephalon within 2 weeks [7,13]. Thus, despite their different origins, various waves of OPCs seem to be functionally redundant. However, many data also suggest functional heterogeneity in OPCs [14]. Migration of OPCs towards the optic nerve can be Nrg1-ErbB4-signaling dependent or independent depending on whether OPCs are generated from the first or second wave [15]. Another example is the Shh-dependent modulation of the generation of OPCs during the first wave, while the other waves are Shh-independent [16]. Independent of their origin, OPCs have different morphologies based on their location [1,17]. OPCs in the white matter have an elongated soma packed tightly between the myelinated axons, with processes extending parallel to the direction of the fibers. OPCs in the gray matter have a star-shaped morphology with several long and thin branches extending in all directions [1]. OPCs can also have different dynamics in terms of proliferation, migration, and differentiation rate [8]. OPCs in the corpus callosum differentiate into oligodendrocytes faster than OPCs in the cortex [8]. OPCs in ventral and dorsal forebrain respond differently to DNA damage by promoting senescence or apoptosis, respectively [18]. A single-cell RNAseq analysis of OPCs isolated from the mouse forebrain at different stages during development (E13.5, P7, and P21) showed that OPCs originating from several regions of the forebrain become transcriptionally equivalent at P7 [19]. However other studies have shown that OPCs are heterogeneous based on their transcriptomes, their surfaceomes (e.g., receptors and ion channels) or electrophysiological properties [17,20,21]. Spitzer and colleagues showed that OPCs start to express voltage-gated ion channels and glutamate receptors differentially throughout development in concordance with their functional states [21]. All these studies demonstrate that OPCs are a diverse population and their diversity changes based on the brain region and age.

A fundamental question is whether the functional diversity of OPCs arises from molecularly defined subpopulations with distinct functions or whether it is a result of their interaction with the environment [1,17]. In this study, we show that expression of *C1ql1*, a gene known for its role in neuronal synapse formation [22,23], marks a subpopulation of OPCs throughout the mouse brain starting around birth. Using genetic ablation, we show that the elimination of this subpopulation leads to lack of OLs in the dorsal forebrain and a massive reduction of myelination. While some OPCs escape *Sox10*-driven ablation and end up compensating the lack of OPCs with time, they cannot rescue OL and myelination deficits. Thus, we identify a molecular signature for a subpopulation of OPCs that generates myelinating OLs during postnatal development of the mouse forebrain.

## Results

### A subpopulation of OPCs is characterized by *C1ql1* expression

Breeding of the *C1ql1^Cre^* knockin mouse model [24] with reporter mouse lines can be used to determine the history of expression of *C1ql1* during brain development. Using the Cre-dependent *R26^Cas9-GFP^* reporter mouse line ([24,25] and Fig 1A), we noticed that, in addition to the expected expression in certain neuronal populations [24], GFP was expressed by a category of cells present throughout the brain at postnatal day 30 (P30) (Fig 1B). These cells did not appear to be neurons, an observation that was confirmed using immunolabeling for NeuN (S1 Fig). While cerebellar granule cells and inferior olivary neurons were both GFP and NeuN positive

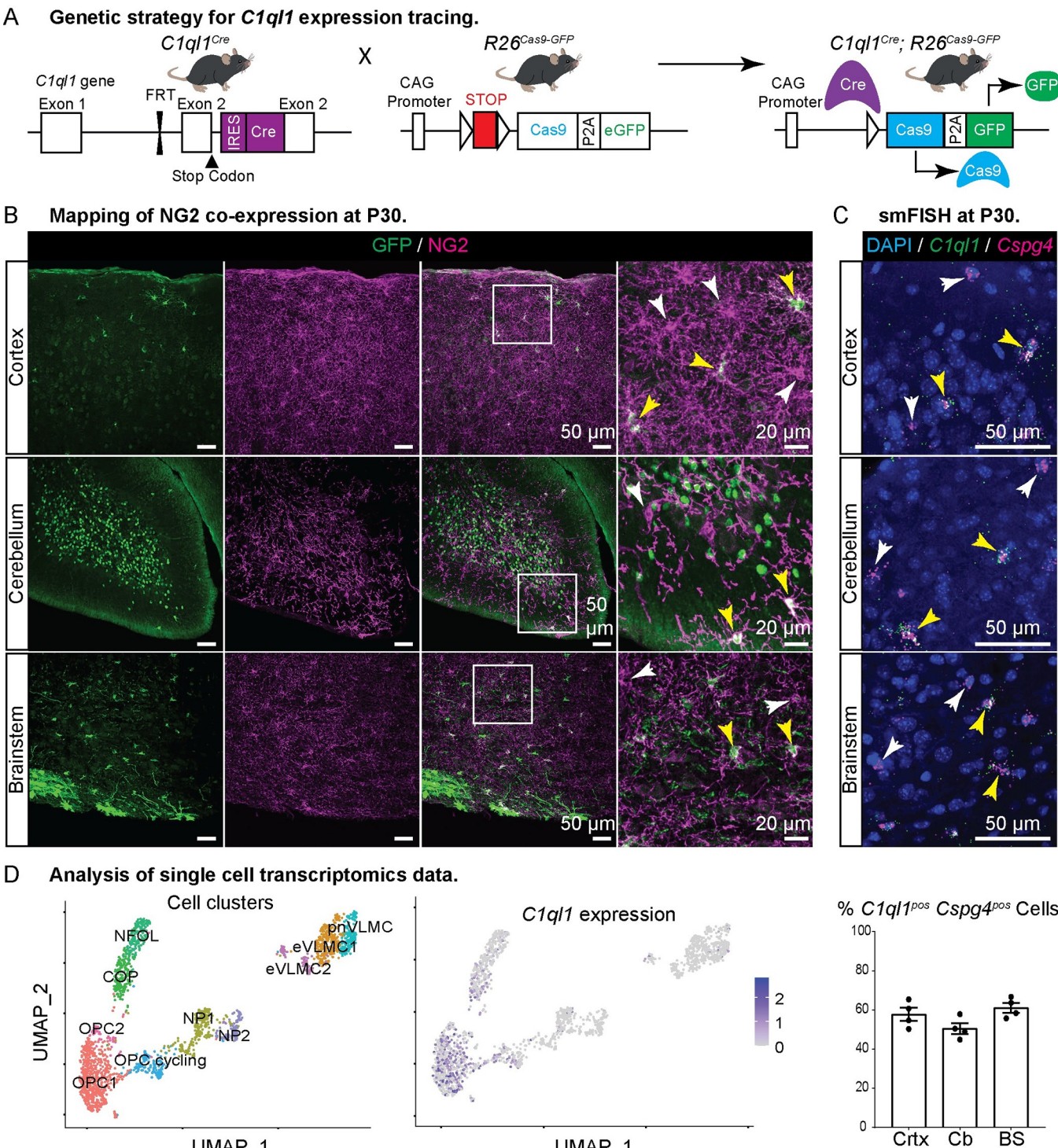

**Fig 1. A subpopulation of oligodendrocyte precursor cells is characterized by *C1ql1* expression.** (A) Diagram of the breeding scheme used to determine the cell types expressing *C1ql1* in the mouse brain. The *C1ql1^{Cre}* mouse line was crossed with the Cre-dependent *R26^{Cas9-GFP}* reporter line. (B) GFP-expressing cells and OPCs were visualized using co-immunolabeling for GFP (green) and NG2 (magenta) respectively, in parasagittal brain sections from *C1ql1^{Cre}; R26^{Cas9-GFP}* animals at postnatal day 30 (P30). In the cerebellum and the brainstem, GFP is strongly expressed by granule cells and inferior olivary neurons, respectively. Scale bars = 50 μm. Higher magnifications of the regions delineated by the white squares show the presence of GFP^{neg} (white arrowhead) and GFP^{pos} (yellow arrowhead) NG2 cells. Scale bars = 20 μm. (C) Duplex smFISH of *C1ql1* and *Cspg4* (coding for NG2) mRNAs was performed in parasagittal sections of the somatosensory cortex, cerebellum, and brainstem from P30 wild-type animals. Both *C1ql1^{neg}* (white arrowhead) and *C1ql1^{pos}* (yellow arrowhead) *Cspg4^{pos}* cells are detected. Scale bars = 50 μm. Bottom: Quantification of the percentage of *C1ql1^{pos} Cspg4^{pos}* cells was performed in the somatosensory cortex (Crtx),

cerebellum (Cb), and brainstem (BS). Data are represented as mean ± SEM. *n* = 4 animals. (D) New analysis of the single-cell transcriptomics raw data from Marques and colleagues [19] reveals the different cell clusters of the oligolineage (left panel) and confirms the expression of *C1ql1* (right panel) in a subset of OPCs and COP cells. COP, committed oligodendrocyte precursors; eVLMC, embryonic vascular and leptomeningeal cells; NFOL, newly formed oligodendrocyte; NP, neural progenitor; OPC, oligodendrocyte precursor cell; pnVLMC, postnatal vascular and leptomeningeal cells; smFISH, single-molecule fluorescence in situ hybridization. The individual numerical values underlying this figure can be found in S1 Data.

as expected (S1 Fig), the other GFP-labeled cells were NeuN negative. Because the morphology of these cells was suggestive of glia, we tested markers for various glial populations. The GFP-positive cells were negative for GFAP showing that they were not astrocytes (S1 Fig). OPCs are characterized by the expression of neural/glial antigen 2 (NG2) and platelet-derived growth factor receptor α (PDGFRa). These markers immediately disappear upon differentiation into OLs [1,9]. Using NG2 immunolabeling, we determined that the non-neuronal cells labeled using the *C1ql1^{Cre}* knockin mouse line were a subpopulation of OPCs: some, but not all, NG2-labeled cells were also GFP-positive (Fig 1B).

Our genetic approach suggests that a subpopulation of OPCs, and/or their progenitors, express *C1ql1*. We performed duplex single-molecule fluorescence in situ hybridization (smFISH) for *C1ql1* and *Cspg4* (coding the marker NG2) on brain sections from P30 mice (Fig 1C). *C1ql1* puncta were observed together with *Cspg4* puncta in a subset of cells throughout the brain. Quantification showed that about 57% (1,476/2,542 cells), 50% (1,122/2,232 cells), and 61% (2,099/3,454 cells) of *Cspg4* expressing cells in the cortex, cerebellum, and brainstem, respectively, express *C1ql1* (Fig 1C). Single-cell transcriptomics of the oligodendrocyte lineage at embryonic (E13.5) and postnatal stages (P7, juvenile, and adult) has previously shown that, besides a subpopulation of cycling OPCs, OPCs form a rather homogenous cell population at the transcriptome level at P7 [19]. Using the raw expression data from these studies [19], we performed our own bioinformatic analysis to determine the pattern of *C1ql1* expression across the different cell clusters constituting the oligodendrocyte lineage. We used a batch correction to eliminate the effects due to the different experimental batches (E13.5 and P7 versus juvenile and adult, cf. S2 Fig), followed by uniform manifold approximation and projection (UMAP) reduction and clustering on shared K-nearest neighbor (SNN) graph. This pipeline led to the identification of 10 clusters including 3 OPC subpopulations (Fig 1D). Two of these subpopulations were the same as the ones found by Marques and colleagues [19] (OPC1 and the cycling OPC subpopulations), while a small third one (OPC2) was defined by the enriched expression of genes associated with the GO terms "mRNA processing" and "mRNA splicing," suggesting a specific metabolism of RNA. *C1ql1*-expressing cells were found in all 3 OPC clusters as well as in a subset of committed OPCs (COPs) that are differentiating into oligodendrocytes (Fig 1D). *C1ql1* expression was almost undetected in newly formed oligodendrocytes (NFOL).

Thus, genetic and transcriptomic data altogether demonstrate that a subpopulation of OPCs expresses *C1ql1* in the juvenile brain and raise the question as to whether this subpopulation of OPCs has a specific role in the brain.

## Ablation of *C1ql1*-expressing OPCs is not compensated during postnatal development

In the adult mouse brain, OPCs maintain their homeostasis by local proliferation [26]. Ablation, differentiation, or apoptosis of OPCs leads to proliferation and migration of the adjacent cells to replace the missing ones [26]. OPCs that are generated in different waves show functional redundancy, as ablation of one wave using *Sox10*-driven expression of diphtheria toxin (DTA) is compensated by the other 2 waves in the cortex during the first 2 postnatal weeks [12]. Even ablation of all 3 waves upon their generation during embryonic development and

birth induces repopulation by the OPCs from other regions of the brain (e.g., diencephalon) within 15 days [7,13], showing the strong capacity of OPCs for sensing OPC disappearance and compensating for it [12,13,26]. We used the previously established *Sox10*-DTA based ablation method to test whether *C1ql1*-expressing OPCs (*C1ql1$^{pos}$* OPCs) are functionally redundant with other OPCs. For this, the *C1ql1$^{Cre}$* mouse line was crossed with the *Sox10$^{DTA}$* line expressing DTA in a Cre-dependent manner [12] (Fig 2A). The *Sox10* promoter is specifically active in all the oligodendroglial lineage and drives GFP expression in more than 98% of the oligolineage cells in the *Sox10$^{DTA}$* line [12]. Accordingly, in the absence of Cre *(C1ql1$^{wt}$; Sox10$^{DTA}$* mice), the vast majority of the oligolineage cells, OPCs, COPs, and OLs, are GFP positive (Fig 2B). The expression of the Cre recombinase using the *C1ql1$^{Cre}$* locus leads to loss of GFP, induction of DTA expression, and death of the Cre-expressing cells in the oligolineage (Fig 2A). *C1ql1* is not expressed in progenitors (Fig 1D), and thus, expression of Cre does not affect the generation of *C1ql1*-expressing neurons. The *C1ql1$^{Cre}$; Sox10$^{DTA}$* mice only rarely survived beyond 3 to 4 postnatal weeks. Because *C1ql1* and *Sox10* are co-expressed in various cell types of the enteric nervous system (http://mousebrain.org/adolescent/genesearch.html), this lethality might be due to its dysfunction, as has been described in other mutants with enteric system dysfunction [27]. All the animals survived until P16 without any major visually detectable behavior or motor deficit. Morphological analysis showed a massive reduction in the number of GFP-expressing cells all over the brain, and in particular in the forebrain, after ablation of *C1ql1*-expressing OPCs (*C1ql1$^{Cre}$; Sox10$^{DTA}$*) compared to controls (*C1ql1$^{wt}$; Sox10$^{DTA}$*, Fig 2B–2D). Since the development of oligolineage cells has been well studied in the forebrain, particularly in the corpus callosum and the cortex [12], we selected the central and lateral parts of the corpus callosum (CC-center and CC-lateral), as well as the retrosplenial area of the cortex for detailed analysis (Fig 2C and 2D). The number of *Sox10*-GFP positive (GFP$^{pos}$) cells is reduced by 66%, 85%, and 46% in these 3 regions, respectively, upon *C1ql1$^{Cre}$*-induced ablation (Fig 2C; mean ± SEM in the control and ablation conditions: CC-center = 41.2 ± 2.1 versus 13.9 ± 1.2 × 10$^4$ cells/mm$^3$; CC-lateral = 39.1 ± 3.0 versus 5.7 ± 0.6 × 10$^4$ cells/mm$^3$; cortex = 10.0 ± 1.5 versus 5.4 ± 0.5 × 10$^4$ cells/mm$^3$). Since GFP labels nearly the entire oligolineage cells in the *Sox10$^{DTA}$* mouse model [12], the OPC marker NG2 [1,9] was used to quantify the OPC subpopulation (NG2$^{pos}$ GFP$^{pos}$ cells). A 57%, 80%, and 32% reduction in the density of NG2$^{pos}$ GFP$^{pos}$ cells was found in the center and lateral parts of the corpus callosum, and cortex, respectively (CC-center: 25.0 ± 4.1 versus 10.7 ± 1.0; CC-lateral: 24.3 ± 3.2 versus 4.8 ± 0.6 × 10$^4$ cells/mm$^3$; cortex: 7.4 ± 1.6 versus 5.0 ± 0.5 × 10$^4$ cells/mm$^3$). In comparison with NG2 staining, immunolabeling for PDGFRa, another marker of OPCs [1,9], is more concentrated on the somata of OPCs and allows easier quantification of individual OPCs (Fig 2C and 2D). These quantifications also showed a reduction in the density of PDGFRa$^{pos}$ GFP$^{pos}$ cells in all 3 regions at P16 (CC-central: 74% decrease, mean ± SEM = 20.1 ± 1.2 versus 5.3 ± 0.7 × 10$^4$ cells/mm$^3$; CC-lateral: 79%; 26.5 ± 3.3 versus 5.7 ± 0.4 × 10$^4$ cells/mm$^3$; cortex: 29%; 7.9 ± 0.8 versus 5.6 ± 0.4 × 10$^4$ cells/mm$^3$). Altogether, using the 2 most common markers of OPCs, NG2 and PDGFRa, these results confirm that DTA expression driven by the *C1ql1$^{Cre}$* knockin line leads to a massive disappearance of OPCs in several regions of the forebrain. They also suggested that, in contrast to what has been reported when ablating OPCs from different waves [12,13], remaining OPCs cannot compensate for the loss of *C1ql1*-expressing OPCs during postnatal development.

### *C1ql1* expression in OPCs follows region-specific time-courses during postnatal development

Lack of compensation of OPC loss by P16 in *C1ql1$^{Cre}$; Sox10$^{DTA}$* brains could be due to a reduced amount of time since ablation compared to previously published results [12,13,28].

**Fig 2. Ablation of *C1ql1*-expressing OPCs is not compensated during postnatal development.** (A) Diagram of the breeding scheme used to ablate *C1ql1*-expressing oligolineage cells in the mouse brain via *Sox10*-driven expression of diphtheria toxin (DTA). The *C1ql1^Cre* mouse line was crossed with the Cre-dependent *Sox10^DTA* mice. (B) Parasagittal and coronal sections from P16 brains of control (*C1ql1^wt; Sox10^DTA*) and *C1ql1^pos* OPC-ablated (*C1ql1^Cre; Sox10^DTA*) mice. Direct GFP fluorescence and Hoechst staining are shown. Scale bars = 1 mm. Numbers indicate the regions of the corpus callosum (1: CC-center, 2: CC-lateral) and cortex 3: used for quantifications. (C) OPCs were immunolabeled for NG2 (magenta) in coronal sections from P16 brains of the control and *C1ql1*-ablated conditions. Both NG2^pos (yellow arrowhead) and NG2^neg (white arrowhead) GFP^pos cells are detected in the center and lateral part of corpus callosum (CC) as well as the cortex. Scale bars = 100 μm. The mean densities of both NG2^pos GFP^pos (OPCs) and NG2^neg GFP^pos oligolineage cells are

reduced in *C1ql1$^{Cre}$; Sox10$^{DTA}$* brains. Data are represented as mean ± SEM. *C1ql1$^{wt}$; Sox10$^{DTA}$*: *n* = 4 animals. *C1ql1$^{Cre}$; Sox10$^{DTA}$*: *n* = 7 animals. Unpaired *t* test with Welch's corrections. ns = not significant. (D) OPCs were immunolabeled for PDGFRa (magenta) in coronal sections from P16 brains of the control and *C1ql1*-ablated mice. Both PDGFRa$^{pos}$ (yellow arrowhead) and PDGFRa$^{neg}$ (white arrowhead) GFP$^{pos}$ cells are detected. Scale bars = 100 μm. Quantification confirms the reduced density of both PDGFRa$^{pos}$ GFP$^{pos}$ (OPCs) and PDGFRa$^{neg}$ GFP$^{pos}$ oligolineage cells upon ablation of *C1ql1*-expressing *Sox10$^{pos}$* cells. Data are represented as mean ± SEM. *C1ql1$^{wt}$; Sox10$^{DTA}$*: *n* = 6 animals. *C1ql1$^{Cre}$; Sox10$^{DTA}$*: *n* = 10 animals. Unpaired *t* test with Welch's corrections. The individual numerical values underlying this figure can be found in S1 Data. OPC, oligodendrocyte precursor cell; PDGFRa, platelet-derived growth factor receptor α.

Furthermore, the magnitude of OPC loss at P16 is bigger in the corpus callosum than in the cortex in the *C1ql1$^{Cre}$; Sox10$^{DTA}$* mice, suggesting regional differences. We analyzed the timing of expression of *C1ql1* in OPCs during postnatal brain development. Analysis of previously published data of single-cell transcriptomics of oligolineage cells [19] shows a negligible expression of *C1ql1* at E13.5, while a considerable number of OPCs express *C1ql1* already by P7 and in juvenile animals (S3A Fig). This suggests that *C1ql1* expression is induced during the first postnatal week in a subpopulation of OPCs in the forebrain. We performed duplex smFISH for *C1ql1* and *Cspg4* on brain sections from wild-type mouse at P7 and P15 (Fig 3A). Similar to our observations at P30 (Fig 1C) and in accordance with the single-cell transcriptomic data (S3A Fig), *C1ql1* labeling was detected in a subgroup of *Cspg4$^{pos}$* cells already by P7 (Fig 3A). Quantification demonstrated that 72% (63,626/88,314 cells) and 68% (77,714/113,512 cells) of *Cspg4$^{pos}$* cells in the cortex express *C1ql1* at P7 and P15, respectively, a percentage higher by about 10% than the percentage quantified at P30 (Fig 1C).

How early does *C1ql1* expression appear in OPCs? smFISH showed the presence of *C1ql1* mRNA in a subgroup of *Cspg4$^{pos}$* cells already by P1 in the cortex (S3B Fig). However, due in particular to the high density of cells in the forebrain at that age, accurate quantification was not possible. To assess the regional differences in the timing of *C1ql1* expression, we thus used a genetic approach and crossed the *C1ql1$^{Cre}$* line with the Cre-dependent *R26$^{R-EYFP}$* reporter line [29]. Co-immunolabeling for YFP, PDGFRa, and NG2 was performed on coronal forebrain slices of P1-P2 *C1ql1$^{Cre}$; R26$^{R-EYFP/WT}$* mice (S3C Fig). About 40% of OPCs expressed YFP in both regions of the corpus callosum by P1-P2 while only 10% in the cortex (S3C Fig; mean ± SEM = 42.4 ± 8.3% versus 35.6 ± 4.1% versus 10.5 ± 0.6% in the central part, lateral part of corpus callosum, and the cortex, respectively). Altogether, our results show that the timing of appearance of *C1ql1* expression in OPCs is region-dependent, with earlier expression in the corpus callosum than in the cortex. In the cortex, this expression appears around birth and increases during the 2 first postnatal weeks of development.

## Loss of *C1ql1*-expressing OPCs is compensated by cells escaping *Sox10*-driven ablation

Does the different time-course of *C1ql1* expression in OPCs across forebrain regions translate into a different time-course of OPC loss in *C1ql1$^{Cre}$; Sox10$^{DTA}$* mice? In the corpus callosum, the percentage of OPCs expressing *C1ql1* at P16 (roughly 70%) is in the range of the OPC loss observed in *C1ql1$^{Cre}$; Sox10$^{DTA}$* mice (70% to 80%, Fig 2). In the cortex, the density of OPC is reduced by only 30% at the same age. Quantification did not reveal changes in the density of oligolineage (GFP$^{pos}$) cells in the brains of *C1ql1$^{Cre}$; Sox10$^{DTA}$* mice at P0 (S4A Fig; 19.4 ± 1.1 versus 23.7 ± 4.5 in CC-center, 32.1 ± 1.9 versus 25.7 ± 3.0 in CC-lateral, and 18.5 ± 1.0 versus 17.7 ± 1.2 × 10$^4$ cells/mm$^3$ in the cortex). At this age, the density of OPCs (PDGFRa$^{pos}$ GFP$^{pos}$ cells) was reduced in the lateral part of the corpus callosum (S4A Fig; 34% decrease; mean ± SEM = 12.9 ± 0.5 versus 8.4 ± 1.4 × 10$^4$ cells/mm$^3$), but not in the center part or in the cortex. At P7, the density of oligolineage cells is already decreased by 40% to 50% in the corpus callosum, while no change is detected in the cortex (S4B Fig, CC-Center:

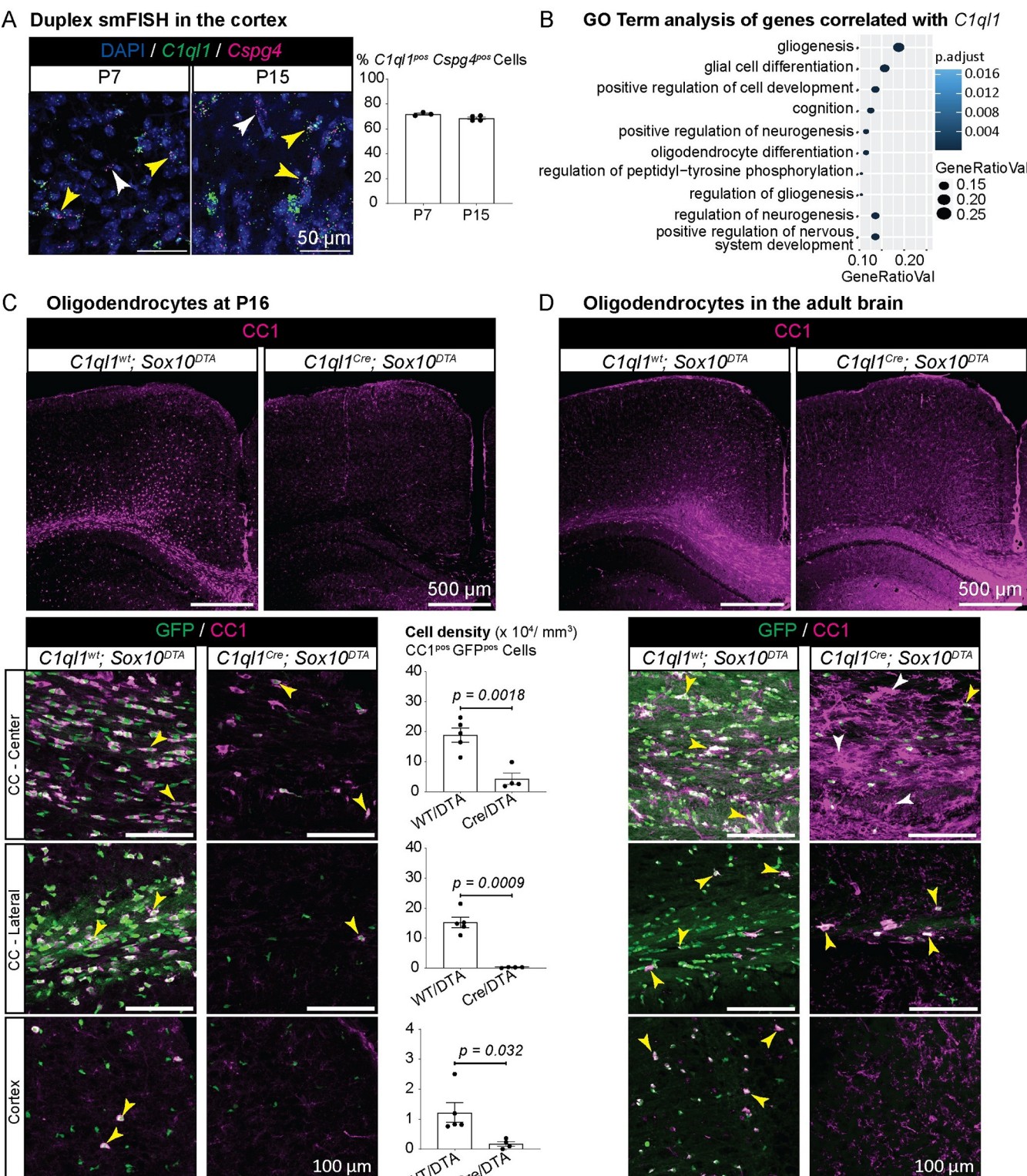

**Fig 3. Lack of oligodendrocytes after DTA-mediated ablation of *C1ql1*-expressing OPCs.** (A) Duplex smFISH of *C1ql1* and *Cspg4* mRNAs was performed in coronal sections from the cortex of P7 and P15 wild-type mice. Both *C1ql1^neg^* (white arrowhead) and *C1ql1^pos^* (yellow arrowhead) *Cspg4^pos^* cells are observed. Scale bars = 50 μm. Quantification of the percentage of *C1ql1*-expressing *Cspg4^pos^* cells was performed in the cortex of P7 and P15 animals. Data are represented as mean ± SEM. P7: *n* = 3 animals. P15: *n* = 4 animals. Scale bars = 50 μm. (B) GO analysis shows the biological functions of the genes enriched in *C1ql1*-expressing OPCs compared to other OPCs (reanalysis of raw data from Marques and colleagues [19]). (C) Coronal sections from *C1ql1^wt^; Sox10^DTA^*

and *C1ql1^{Cre}; Sox10^{DTA}* brains at P16 were immunolabeled for CC1, a marker of oligodendrocytes (magenta). Scale bars = 500 μm. Higher magnification of the coronal sections in the center and lateral parts of CC and cortex shows the dramatic reduction in the density of OLs (CC1^{pos} GFP^{pos} cells, yellow arrowhead) in *C1ql1^{Cre}; Sox10^{DTA}* mice at P16. Scale bars = 100 μm. Quantification shows a massive decrease in OL density in the *C1ql1^{pos}* OPC-ablated brains. Data are represented as mean ± SEM. *C1ql1^{wt}; Sox10^{DTA}*: n = 5 animals. *C1ql1^{Cre}; Sox10^{DTA}*: n = 4 animals. Unpaired *t* test with Welch's corrections. ns = not significant. (D) Coronal sections from *C1ql1^{wt}; Sox10^{DTA}* and *C1ql1^{Cre}; Sox10^{DTA}* cortex from adult mice were immunolabeled for CC1. Scale bars = 500 μm. Higher magnification shows rare CC1^{pos} GFP^{pos} *Sox10* lineage-derived OLs in the corpus callosum of *C1ql1^{Cre}; Sox10^{DTA}* adult brains, and repopulation by GFP^{neg} OLs (GFP^{neg} CC1^{pos} cells, white arrowhead) only in the center of the corpus callosum. In the cortex, cells that do not seem to be OLs are CC1 positive (probably astrocytes as seen in [34]). Scale bars = 100 μm. The individual numerical values underlying this figure can be found in S1 Data. GO, Gene Ontology; OPC, oligodendrocyte precursor cell; smFISH, single-molecule fluorescence in situ hybridization.

mean ± SEM = 32.7 ± 1.9 in the control versus 16.3 ± 1.5 × $10^4$ cells/mm$^3$; CC-lateral: 45.9 ± 2.5 versus 26.6 ± 2.5 × $10^4$ cells/mm$^3$; cortex: 13.3 ± 1.0 versus 15.9 ± 3.2 × $10^4$ cells/mm$^3$). The density of OPCs (PDGFRa^{pos} GFP^{pos} cells) was found to be decreased by about 55% in the center and lateral corpus callosum, while no change was detected in the cortex (S4B Fig; mean ± SEM: CC-center = 17.1 ± 1.2 versus 7.6 ± 2.5 × $10^4$ cells/mm$^3$; CC-lateral = 14.9 ± 1.3 versus 6.9 ± 2.1 × $10^4$ cells/mm$^3$; cortex = 7.1 ± 0.4 versus 6.9 ± 2.5 × $10^4$ cells/mm$^3$). These results show that the time-course of OPC loss in the 3 analyzed regions correlates with the time-course of appearance of *C1ql1* in OPCs, with the cortex being affected later than the corpus callosum.

In the cortex, OPC loss is not yet observed at the end of the first postnatal week, despite the presence of about 10% *C1ql1^{pos}* OPCs already at P1-P2 and the steep increase of this percentage during the first week. The slower time-course of OPC loss in the cortex could be due to increased proliferation of remaining OPCs. To assess this, we used immunolabeling for Ki67, a marker for proliferative cells [30] (S5 Fig). No significant change in the percentage of proliferative oligolineage cells (Ki67^{pos} GFP^{pos} cells) was detected in the cortex at neither P0 nor P7 (mean ± SEM = 22.3 (2,308/10,692 cells) ± 2.5% versus 26.3 (2,799/10,643 cells) ± 1.8% at P0 and 5.3 (224/4,673 cells) ± 1.7 versus 7.7 (260/4,312 cells) ± 2.7% at P7). Another factor that could slow down OPC loss in this region during the first postnatal week could be the migration of newly formed OPCs, which ends only by P10 in the dorsal forebrain [12]. Regardless of these regional differences, the ablation of OPCs in *C1ql1^{Cre}; Sox10^{DTA}* mice does not seem to be compensated by the remaining OPCs during postnatal development, since a significant decrease in OPCs is detected by P16 in all 3 regions.

Could compensation occur with longer recovery time? Rarely, some *C1ql1^{Cre}; Sox10^{DTA}* mice survived into adulthood (>8 weeks). Morphological analysis showed a global massive decrease in the number of GFP-expressing cells in the forebrain of these mice compared to *C1ql1^{wt}; Sox10^{DTA}* controls, similarly to our observations at P16 (S6A Fig). Immunolabeling for NG2 and PDGFRa showed only rare *Sox10*-GFP^{pos} PDGFRa^{pos} NG2^{pos} cells and the appearance of GFP^{neg} PDGFRa^{pos} NG2^{pos} cells in the cortex and the central and lateral parts of the corpus callosum (S6B Fig). This means that a compensation of OPC loss can occur after many weeks, but from cells in which the promoter of *Sox10* has low or no activity (less than 2% of the cells [12,28]) and thus escape *Sox10^{DTA}*-mediated ablation in the presence of Cre recombinase. These cells could be derived locally or from neural progenitor cells in the ventricular-subventricular zone, as observed several weeks after ablation by pharmacological and genetic tools of all OPCs in the adult mouse brain [28].

## Ablation of *C1ql1*-expressing OPCs leads to lack of oligodendrocytes

*C1ql1*-expressing cells are present in all OPC clusters defined by single-cell transcriptomics (Fig 1D). This implies that instead of defining a genetically distinct subgroup, *C1ql1* is a marker of a specific transitional state and/or a specific function within the OPC subpopulation.

We performed overrepresentation analysis of the Gene Ontology (GO) terms associated with $C1ql1^{pos}$ OPCs compared to other OPCs in single-cell transcriptomic data (our reanalysis of data from [19], Figs 1D, 3B, S7A and S7B). "Gliogenesis" and "glial cell differentiation" were the 2 most significantly overrepresented terms (gene ratio = 0.19 and 0.15, respectively) in $C1ql1^{pos}$ OPCs (Figs 3B and S7A), suggesting that $C1ql1^{pos}$ OPCs might already be committed to differentiation. Indeed, $C1ql1$ expression is also detected in some committed oligodendrocyte precursors (COPs) but disappears in NFOLs (Fig 1D). Other markers have been previously found to label a specific transition stage of OPC differentiation in OLs [31–33]. The long noncoding RNA $Pcdh17it$ labels newly born immature OLs (as confirmed by our analysis, Figs 1D and S7B), while $Gpr17$ and $Bcas1$ are markers of COPs [32,33] (Figs 1D and S7B). $C1ql1^{pos}$ cells do not overlap with cells expressing $Gpr17$, $Bcas1$, and $Pcdh17it$ markers to a significant extent, while many $C1ql1^{pos}$ cells coexpress $Cspg4$ (S7B Fig). Thus, $C1ql1$ expression is not another marker of COPs and NFOLs, but rather marks a specific state in a subpopulation of OPCs that is ready for differentiation into oligodendrocytes. Indeed, quantification of the number of $GFP^{pos}$ cells after ablation of $C1ql1$-expressing OPCs showed a massive decrease not only in $NG2^{pos}$ $GFP^{pos}$ cells, but also in $NG2^{neg}$ $GFP^{pos}$ cells (Fig 2).

Because GFP expression is driven by the $Sox10$ promoter, this suggested that DTA ablation in $C1ql1^{pos}$ OPCs results in the reduction not only in OPC numbers but also in COPs and oligodendrocytes. Indeed, at P16 $NG2^{neg}$ $GFP^{pos}$ cells are completely absent in the cortex and corpus callosum. Using immunolabeling of P16 forebrain coronal slices for CC1, a marker of oligodendrocytes, we confirmed the massive reduction in OL numbers after ablation of $C1ql1$-expressing OPCs, with an almost complete absence of $CC1^{pos}$ cells in the cortex (Fig 3C). Quantification demonstrated a 77%, 98%, and 83% reduction in the number of OLs ($CC1^{pos}$ $GFP^{pos}$ cells) in the center (mean ± SEM = 18.9 ± 2.3 versus 4.2 ± 1.9 × $10^4$ cells/mm$^3$) and lateral (mean ± SEM = 15.2 ± 1.7 versus 0.2 ± 0.0 × $10^4$ cells/mm$^3$) corpus callosum, and in the cortex (mean ± SEM = 1.2 ± 0.3 versus 0.2 ± 0.1 × $10^4$ cells/mm$^3$), respectively. In the adult, several weeks after ablation of $C1ql1$-expressing OPCs, only very few $GFP^{pos}$ $CC1^{pos}$ cells were visible in the corpus callosum and cortex of $C1ql1^{Cre}$; $Sox10^{DTA}$ forebrain (Fig 3D). $GFP^{neg}$ $CC1^{pos}$ OLs were present only in the center of the corpus callosum. Of note, in this region, CC1 cellular expression was also up-regulated compared to the level observed in control animals. In the cortex, cells that do not seem to be OLs are CC1 positive (probably astrocytes as seen in [34]). Thus, despite the repopulation by $GFP^{neg}$ OPCs in the adult $C1ql1^{Cre}$; $Sox10^{DTA}$ brain (S6B Fig), there is no generation of OLs other than in the center of the corpus callosum. These results show that $C1ql1$-expressing OPCs are essential for the generation and/or survival of OLs in the brain.

## Ablation of *C1ql1*-expressing OPCs leads to lack of myelination

OL generation starts around birth in mice and continues throughout life [10]. Myelination starts from the end of the first postnatal week and increases rapidly to reach its maximum level by the end of the second week [10]. This intense period of myelination coincides with the timing of the defects in generation of OLs observed after ablation of $C1ql1$-expressing OPCs. Immunolabeling for the early marker of myelination, myelin basic protein (MBP), revealed a dramatic decrease in MBP in different regions of the forebrain at P16 after ablation of $C1ql1$-expressing OPCs (Fig 4A). MBP labeling was almost completely absent in the lateral corpus callosum (84% of reduction; mean thickness ± SEM = 144.6 ± 16.8 versus 23.6 ± 4.9 μm) and in the cortex (81% of reduction; mean raw integrated density ± SEM = 445.8 ± 74.2 versus 86.4 ± 7.3). In the center of the corpus callosum, a slight but not statistically significant, decrease was observed in the thickness of the region

**Fig 4. Lack of myelination in the forebrain after genetic ablation of *C1ql1*-expressing OPCs.** (A) Immunolabeling of coronal P16 brain sections for MBP (magenta) illustrates the almost total absence of myelination in the forebrain upon genetic ablation of *C1ql1*-expressing OPCs, without gross morphological changes. Scale bars = 1 mm. (B) Myelinated fibers (immunolabeled for MBP, magenta) were visualized in coronal sections of the center and lateral part of the corpus callosum and cortex at P16. Quantification of the thickness of myelinated regions in the corpus callosum (white dashed lines) and the intensity of MBP in the cortex shows an almost complete absence of MBP in CC-lateral and cortex. Data are represented as mean ± SEM. *C1ql1^{wt}; Sox10^{DTA}*: $n$ = 4 animals. *C1ql1^{Cre}; Sox10^{DTA}*: $n$ = 7–10 animals. Unpaired *t* test with Welch's corrections. Raw Int Den: Raw integrated density. Scale bars = 100 μm. (C) Staining for FluoroMyelin Red (green) in coronal sections shows the decrease in compact myelin in the corpus callosum of P16 mice after ablation of *C1ql1^{pos}* OPCs. Compact myelin was not present in the cortex of control mice suggesting that at P16, myelination is still immature in this region. Scale bars = 100 μm. The individual numerical values underlying this figure can be found in S1 Data. MBP, myelin basic protein; OPC, oligodendrocyte precursor cell.

containing MBP labeled axon bundles (mean ± SEM = 320.9 ± 16.8 versus 293.6 ± 12.8 μm). Immunolabeling for 2′,3′-cyclic nucleotide 3′-phosphodiesterase (CNP), another marker of early events of myelin formation [35], confirmed this phenotype. A 92% and 51% decrease in CNP labeled fibers was observed in the lateral part of the corpus callosum (mean ± SEM = 145.8 ± 12.9 versus 11.8 ± 9.1 μm) and cortex (mean ± SEM = 902.8 ± 135.2 versus 442.7 ± 39.9), respectively (S8A Fig). In the center of the corpus callosum, the thickness of CNP labeling was reduced by 26% (mean ± SEM = 409.4 ± 26.5 versus 304 ± 24.8 μm) (S8A Fig). In the central part of the corpus callosum, both MBP and CNP intensity is increased in *C1ql1^{pos}* OPCs-ablated condition compared to controls. Together with the increased expression of CC1, this observation suggests a specific reactive state of the OLs produced by GFP^{neg} OPCs in compensation for the loss of *C1ql1^{pos}* OPCs. Staining for FluoroMyelin, a marker for compact myelin [36], at P16 shows the presence of compact myelinated sheaths in the center and lateral part of the corpus callosum, but not in the cortex suggesting that myelination is not mature in this region at this stage (Fig 4C). After ablation of *C1ql1^{pos}* OPCs, compact myelin was completely absent in the lateral part of the corpus callosum. A dramatic reduction in the labeling was observed in the central part, with the remaining stained fibers not forming a clear bundle. To rule out an indirect effect of eliminating *C1ql1*-expressing OPCs on axonogenesis, we assessed the integrity of axons and neuronal fibers using immunolabeling for the neurofilament protein (NF) [9,37] (S8B Fig). No difference was observed in the thickness of fiber tracts in the center of the corpus callosum (mean ± SEM = 326.5 ± 12.3 versus 292.7 ± 17.1 μm) and the intensity of NF labeling in the cortex (mean ± SEM = 689.2 ± 251.5 versus 830.7 ± 216.4). A 19% reduction in the thickness of the lateral part of the corpus callosum was detected (mean ± SEM = 129.1 ± 7.8 versus 104.9 ± 2.8 μm). Given the much higher reduction in MBP staining (84%), this reduced thickness is most likely due to increased compaction of axons rather than to deficits in axonogenesis. In adult *C1ql1^{Cre}; Sox10^{DTA}* mice, immunolabeling for MBP and CNP demonstrated the persistent absence of myelin in the cortex and lateral corpus callosum (S9A Fig). At this stage, staining with FluoroMyelin and imaging with spectral confocal reflectance microscopy (SCoRe) both showed the clear presence of bundles of compact myelin in the corpus callosum and cortex in control conditions (S9B Fig). After ablation of *C1ql1*-expressing OPCs, compact myelin was visible in the central part of the corpus callosum, but greatly reduced in the lateral part and completely absent in the cortex. Overall, our results show a major lack of myelination resulting from the ablation of *C1ql1*-expressing OPCs during perinatal and postnatal development that persists into adulthood in various parts of the forebrain.

## Discussion

Whether the various functions of OPCs are related to their molecular heterogeneity or are simply the result of interactions with their local environment is unresolved [2,14]. In this study, we identified a subpopulation of OPCs in the mouse brain defined by the expression of *C1ql1*, a gene coding for a synaptic protein. This subpopulation appears starting around birth. Its ablation is not compensated by remaining OPCs in the dorsal forebrain during the first 2 weeks of postnatal development and leads to OL and myelination deficiency. With time, GFP^{neg} OPCs in which the activity of the promoter of *Sox10* is likely very low or absent, repopulate the forebrain. However, they are unable to compensate for the OL and myelination deficits in many brain regions such as the cortex that remains mostly unmyelinated.

Single-cell RNA sequencing of cells from the oligodendroglia lineage during embryonic and postnatal development previously showed that OPCs with different origins converge transcriptionally at P7 to form a homogeneous population [19]. However, the idea that OPCs are equivalent has been challenged, either by single-cell transcriptomic studies of adult OPCs or by functional analysis [18,20]. Here, we present new evidence for the molecular and functional diversity of OPCs. The expression of *C1ql1* appears around birth in a subset of OPCs, with a time course that differs depending on the brain region. In the cortex, the percentage of *C1ql1*-expressing OPCs reaches its maximum number at P7 and then decreases gradually until P30. Our results also show that the capacity for compensation of OPC loss during postnatal development is different between brain regions. It has been reported that 4% of oligodendroglia in the corpus callosum are *Sox10*-negative [38]. A low percentage of *Sox10*-expressing cells do not express GFP in *Sox10^DTA* animals [12,28], due to low levels of activity of the *Sox10* promoter. Both types of cells could give rise to GFP^neg OPCs in the adult *C1ql1^Cre; Sox10^DTA* forebrain. The transient deficit in OL generation and myelination in the center of the corpus callosum after ablation of *C1ql1*-expressing OPCs is rescued with time by GFP^neg OPCs. However, in the lateral part of the corpus callosum and in the cortex, these GFP^neg OPCs, while they also repopulate these regions, are unable to rescue the OL and myelination deficits. Furthermore, the fact that *C1ql1*-expressing OPC ablation can only be rescued by GFP^neg oligolineage cells suggests that *C1ql1* marks an indispensable transient state for OL generation and myelination during normal development.

*C1ql1*-expressing OPCs are found in both white (such as brainstem) and gray matters (e.g., cerebral cortex) in various regions of the mouse brain, and *C1ql1* expression is not restricted to a particular cluster of OPCs. This suggests that the transient state marked by *C1ql1* expression in OPCs might be induced by interaction of OPCs with the microenvironment or other cells in the tissue. The interaction of OPCs with different types of cells in the brain, such as microglia [2,39] or endothelial cells [40], has been shown to be necessary not only for the function of those cells, but also for proper development of OPCs themselves [2,39,40]. C1QL1 is a synaptic molecule involved in excitatory synapse formation and maintenance between neurons [22,23,41]. Its receptor is the postsynaptic receptor brain angiogenesis inhibitor 3 (BAI3) found in neurons [22,23,41] but also expressed by some glial cells [42]. OPCs could thus interact with their local environment, neurons, and/or other glial cells, via C1QL1/BAI3 signaling. OPCs receive synaptic inputs from both excitatory and inhibitory neurons [43–45]. These synapses have been recently shown to be important for myelination and OPC development in zebrafish [46]. An intriguing hypothesis is that C1QL1/BAI3 signaling could participate in the regulation of neuron-OPC synapses and their control of myelination.

What is the state marked by *C1ql1* expression in OPCs? Interestingly, we found in our analysis of single-cell transcriptomics that expression of the gene *Gos2* is enriched in *C1ql1*-expressing OPCs compared to other OPCs (fold change 1.2; adjusted *p*-value <0.001). *Gos2* is a tumor suppressor gene and might regulate quiescence in proliferative hematopoietic stem cells [47]. In the zebrafish spinal cord, OPCs have been classified into 2 subpopulations based on several characteristics, including their location, function, calcium activity, and dynamics [17]. It has been proposed that these 2 functionally distinct subpopulations of OPCs might represent different states of cell fate determination [17]. In this model, certain OPCs can divide to generate both proliferative OPCs and OPCs which differentiate into OLs [17]. *C1ql1* could mark a similar OPC state before the proliferation and production of OPCs primed for differentiation into OLs. This model would explain why we do not detect an increase in the proliferation of the remaining OPCs upon the ablation of *C1ql1*-expressing ones: any OPC trying to divide to produce OL would go through this *C1ql1*-marked state and die. Neuronal activity is important for controlling myelination [48,49] and might induce this specific state marked by *C1ql1* expression in OPCs.

Our discovery might have implications for several types of diseases. C1QL1 has been proposed to have a proliferative and tumor-promoting function in 2 different cancers, including one inside the brain [50,51], and contributes to the regulation of the ovarian follicle reserve [52]. Thus, C1QL1 could control proliferation and/or differentiation of OPCs, in various contexts where an equilibrium between apoptosis and proliferation of oligolineage cells is essential [1,53]. The expression of *C1ql1* is maintained in the adult [20,54], which suggests the importance of *C1ql1*-expressing OPCs in myelination/re-myelination even after the first postnatal month. Oligolineage cells are at the center of understanding the etiology and treatment of demyelination, a pathological condition where the myelin around the axons is lost, and dysmyelination, a condition where the myelin is not formed properly [55]. For example, in multiple sclerosis (MS), a pathology characterized by demyelination, OPCs have been traditionally studied for the development of remyelination strategies. However, a recent study showed that OPCs might be directly involved in the etiology of the disease [56]. Due to their importance in myelination, *C1ql1*-expressing OPCs could be a potential target for therapeutic approaches in this type of disease.

## Materials and methods

All the methods were performed in accordance with the relevant guidelines and regulations.

### Animals

All mice were kept in the authorized animal facilities of CIRB, College de France, under a 12-h light: 12-h dark cycle with water and food supplied ad libitum. All animal protocols and animal facilities were approved by the Comité Régional d'Ethique en Expérimentation Animale (#2001) and the veterinary services (D-75-05-12). All the methods were performed accordingly, and in accordance with the Animals Research: Reporting of In Vivo Experiments (ARRIVE) guidelines and regulations. The *C1ql1$^{Cre}$* mouse model was generated and characterized previously [24]. The Cre-dependent reporter mouse lines *R26$^{Cas9-GFP}$* (B6J.129(B6N)-Gt (ROSA)26Sortm1(CAG-cas9*,-EGFP)Fezh/J, strain #026175) [25] and *R26$^{R-EYFP}$* (B6.129X1-Gt(ROSA)26Sor$^{tm1(EYFP)Cos}$/J, strain #:006148) [29] were received from the Jackson Laboratory. All the lines are maintained on the C57BL/6J background. C57Bl6/J and OF1 mice (Charles River Laboratories, Wilmington, United States of America) were used for the smFISH. *Sox10$^{DTA}$* mice (Sox10-lox-GFP-STOP-lox-DTA, MGI:4999728) [12] were a kind gift from Pr. William D. Richardson (University College London, United Kingdom). They were maintained on a mixed CBA/CaCrl and C57BL/6J background in the original animal facility and were crossed with *C1ql1$^{Cre}$* line upon arrival to CIRB, College de France. In all experiments, littermate mice from both sexes were used as controls for the analysis.

### Single-molecule fluorescent in situ hybridization (smFISH)

Mice at P7, P15, and P30 were perfused intracardially with 4% paraformaldehyde (PFA) phosphate-buffered saline (PBS) solution. Brains were postfixed with the same solution at 4°C for 24 h, transferred to 20% and 30% sucrose/PBS at 4°C sequentially for 24 h, and 30-μm thick sections (P7 and 15: coronally, P30: parasagittally) were obtained using a freezing microtome and stored at −20°C in cryo-preservative solution until use. The smFISH labeling was performed using RNAscope Multiplex Fluorescent Assay kit (Advanced Cell Diagnostics, Newark, USA, cat#323100) according to manufacturer's instructions. Duplex in situ hybridization was performed with the *Cspg4* (ACD, cat#404131) and *C1ql1* RNAscope probes (ACD, cat#465081). The nuclei were labeled with DAPI. Sections were mounted with Prolong Gold Antifade reagent (Invitrogen, cat#P36930). Images were taken using a Zeiss spinning-disk

confocal CSU-W1 microscope with 40× oil objective. A home-made plugin in ImageJ/Fiji was used to detect and quantify the number of individual RNA puncta and their total surface inside each segmented nucleus. The nuclei with a total surface of the RNA puncta of >1.5 $\mu m^2$ were considered positive for that probe.

## Immunofluorescence

Mice at P7, P16, P30, and adult were fixed using intracardiac perfusion of 4% PFA in PBS solution. Brains were extracted, postfixed with the same solution at 4°C for 2 to 4 h, and transferred to 30% sucrose/PBS at 4°C for cryoprotection. 60-μm thick sections were obtained using a freezing microtome and kept in 0.02% $NaN_3$ in PBS solution at 4°C until use. For CC1 (Fig 3B and 3C), Ki67 (S5C Fig), and FluoroMyelin Red staining, as well as SCoRe imaging, 2% PFA in PBS was used for perfusion and postfixation.

To perform immunolabeling, slices were incubated in blocking buffer (4% donkey serum and 1% Triton X-100 in PBS solution) for 1 h at room temperature, followed by incubation with primary antibodies in 1% donkey serum and 1% Triton X-100 in PBS, overnight at 4°C with agitation. Primary antibodies were: GFP (1:1,000), chicken, ab13970, Sigma; NG2 (1:1,000), rabbit, ab5320, Chemicon; APC/CC1 (1:200), mouse, OP80, Calbiochem; PDGFRa (1:500), rat, 14-1401-81, eBioscience; MBP (1:2,000), rat, ab7349, abcam; NF (1:1,000), chicken, ab4680, abcam; GFAP (1:500), mouse, G3893, Sigma; NeuN (1:500), rabbit, ab177487, abcam; CNP (1:500), mouse, C5922-100UL, Sigma; Ki67 (1:1,000), rabbit, ab15580, abcam. The slices were washed 3 times in 1% Triton X-100 in PBS for 10 min, followed by incubation with secondary antibodies (Alexa Fluor 488-, and 568- and 647-labeled donkey/goat anti-mouse, rat, rabbit, or chicken IgGs (H+L); 1:1,000, Invitrogen or Life Technologies) in 1% Triton X-100 in PBS for 2 h at room temperature. Then, sections were washed 3 times in 1% Triton X-100 in PBS for 10 min and incubated for another 10 min at room temperature with Hoechst 33342 (0,2 mg/ml, Sigma, Gothenburg, Sweden, cat#H6024) and 0.4% Triton X-100 in PBS. Sections were mounted using ProLong Gold Antifade Reagent (Invitrogen, cat#P36930).

## FluoroMyelin Red staining

FluoroMyelin Red Fluorescent Myelin Stain was purchased from Thermo Fisher Scientific (Invitrogen, cat#F34652). Coronal slices were incubated with FluoroMyelin Red solution (1:300 in PBS) for 45 min at room temperature, followed by 3 washes of 10 min with PBS. Images were taken using a Zeiss spinning-disk confocal CSU-W1 microscope with 25× oil objective (Z-plane step size: 0.68 μm).

## Spectral confocal reflectance microscopy (SCoRe)

Spectral confocal reflectance microscopy (SCoRe) was performed as described previously [57] using Leica SP5 upright confocal microscope with a water immersion 20× objective. Only 488, 561, and 633 nm lasers were used. The 3 channels were merged together and shown in the figures.

## Image acquisition and analysis

Images for global brain morphology (Figs 2B, 3C, 4A, S6A and S9A) were obtained using a Zeiss Axiozoom V16 macroscope, equipped with a digital camera (AxioCam HRm) using a 16× (pixel size: 4.037 μm), 63× (pixel size: 1.032 μm), or 80× objectives (pixel size: 0.813 μm). Images for NG2, PDGFRa, CC1, Ki67, MBP, CNP staining were acquired using a Zeiss

spinning-disk confocal CSU-W1 microscope with 25× oil objective (Z-plane step size: 0.68 μm). The first 15 z-planes were projected and shown in the figures.

Thickness of the corpus callosum (labeled with MBP, CNP, or NF) was measured manually using "Measure" function in ImageJ/Fiji [58]. For MBP, CNP, or NF labeling in the cortex, the first 15 z-planes of each image were projected using "Z Project" function. An ROI was drawn around layers 4 and 5 of the cortex. Background was subtracted using "Subtract Background." The raw integrated density of the signal was measured using "Measure" function. The value was divided by the area of the ROI in $\mu m^2$ and reported. All the steps were performed in Fiji.

A home-made plugin was used to quantify the number of GFP, NG2, PDGFRa, and CC1 cells. Since NG2 is a chondroitin sulfate proteoglycan which is expressed on the cell surface of OPCs [59], it is difficult to segment the NG2 labeling and count them automatically. Therefore, in this plugin, Hoechst and GFP signals are segmented using the StarDist plugin in Fiji. The volume of the signal is filtered by the minimum and maximum values selected by the user. For GFP signal, there is a threshold for minimum intensity. This minimum intensity is measured manually by the user based on the comparison of the signal of the real GFP versus the background. The detected nuclei colocalized with segmented GFP signals are considered as the OL lineage cells (GFP cells). Then, the plugin dilates the area of each segmented nucleus by 1 μm and measures "meanIntCor" for NG2, PDGFRa, or CC1, based on the following formula:

$$meanIntCor = meanInt - bg$$

"meanInt" is the mean intensity of the signal of interest (NG2, PDGFRa, CC1) for dilated nucleus. "bg" is the mean intensity of the minimum signal in the projection.

Then, a threshold is selected for "meanIntCor" value to count the number of OPCs (NG2 or PDGFRa) or OLs (CC1).

## Single-cell transcriptomic data analysis

Metadata and expression matrix from sequencing of single cells of the oligolineage at E13.5, P7, and juvenile/adult mouse (NCBI GEO Series: GSE95194; Marques and colleagues (2018) [19]) were downloaded as.RDS from "https://cells.ucsc.edu/?ds=oligo-lineage-dev". R version 4.1.0 (2021-05-18) and Seurat_4.1.0 packages were used for analysis. Data were normalized and scaled before running the principal component analysis (PCA). Cell cycle scoring did not show any bias that could impact the PCA analysis. Further analysis (S2 Fig) showed the batch effect between the developmental and juvenile/adult data that were not generated at the same time (Marques and colleagues (2016) [32]). In order to remove the resulting technical variations, we used Harmony (v 0.1.1) to integrate the data. The results matrix was used as input for UMAP and the identification of cell clusters thanks to the SNN graph (FindNeighbors and FindClusters R functions). In order to determine the useful resolution, we used clustree (v_0.5.0) and chose clusters generated at the resolution of 0.8. Overrepresentation and Gene Set Enrichment analyses were performed using clusterProfiler (v4.0.5). Assignment of clusters to cell types was achieved using the database msigdbr (R package v_7.5.1, cell type signature gene sets (M8) from *mus musculus*) and the data from the original Marques and colleagues (2018) [19] analysis. The GO database was used to compare $C1ql1^{pos}$ versus $C1ql1^{neg}$ cells.

## Supporting information

**S1 Fig. GFP is expressed in cells that are not astrocytes or neurons in $C1ql1^{Cre}$; $R26^{Cas9-GFP}$ brains.** GFP-expressing cells, astrocytes, and neurons were visualized using co-immunolabeling for GFP (green), glial fibrillary acidic protein (GFAP), and NeuN, respectively, in sagittal brain sections from $C1ql1^{Cre}$; $R26^{Cas9-GFP}$ animals at P30. The granule cells in the cerebellum

and the inferior olivary neurons in the brainstem are co-labeled with GFP and NeuN as previously described [24]. GFP$^{pos}$ GFAP$^{neg}$ NeuN$^{neg}$ cells (white arrowhead) are observed in all analyzed regions of the brain. Scale bars = 50 μm for cortex, scale bars = 100 μm for cerebellum and brainstem.
(EPS)

**S2 Fig. UMAP analysis of the single-cell transcriptomic raw data from Marques and colleagues [19].** (A) Data that were obtained in 2 separate studies (Juvenile and Adult data from Marques and colleagues (2016) [32] and E13.5 and P7 data from Marques and colleagues (2018) [19]) appeared initially as separate in the UMAP representation. (B) After the integration of the data using Harmony, the batch effect was absent.
(EPS)

**S3 Fig. *C1ql1* is already expressed in some OPCs at birth.** (A) Violin plot of single-cell RNA seq data (from Marques and colleagues [19]) during embryonic and postnatal development shows expression of *C1ql1* as early as P7, but not at E13.5. (B) Duplex smFISH for *C1ql1* and *Cspg4* mRNAs was performed in coronal sections from the dorsal forebrain of P1 wild-type mice. Both *C1ql1$^{neg}$* (white arrowhead) and *C1ql1$^{pos}$* (yellow arrowhead) *Cspg4$^{pos}$* cells are observed. Scale bars = 20 μm. (C) Immunolabeling for YFP (green), PDGFRa (cyan), and NG2 (magenta) in coronal sections from P1-P2 brains of *C1ql1$^{Cre}$*; *R26$^{R-EYFP/WT}$* mice shows both YFP$^{pos}$ (yellow arrowhead) and YFP$^{neg}$ (white arrowhead) PDGFRa$^{pos}$ NG2$^{pos}$ cells in the center and lateral part of corpus callosum (CC) as well as the cortex. Scale bars = 50 μm. Quantification of the percentage of YFP$^{pos}$ OPCs (identified as PDGFRa$^{pos}$ NG2$^{pos}$ cells) was performed in the CC-center (CC-Cent), CC-lateral (CC-Lat), and cortex (Crtx) of P1-2 *C1ql1$^{Cre}$*; *R26$^{R-EYFP/WT}$* mice. Data are represented as mean ± SEM. *n* = 3 animals. The individual numerical values underlying this figure can be found in S1 Data.
(EPS)

**S4 Fig. Loss of OPCs starts during the first postnatal week in the corpus callosum after genetic ablation of *C1ql1*-expressing OPCs.** (A) OPCs were immunostained for PDGFRa (magenta) in coronal sections from P0 mice. PDGFRa$^{pos}$ (yellow arrowhead) and PDGFRa$^{neg}$ (white arrowhead) GFP$^{pos}$ cells are identified in the CC-center, CC-lateral, and the cortex. Scale bars = 50 μm. Quantification of GFP$^{pos}$ (green) and PDGFRa$^{pos}$ GFP$^{pos}$ (magenta) cells shows a decrease in the density of OPCs in CC-lateral, but not in CC-center or in the cortex of *C1ql1$^{Cre}$*; *Sox10$^{DTA}$* mice compared to controls. Data are represented as mean ± SEM. *C1ql1$^{wt}$*; *Sox10$^{DTA}$*: *n* = 3–5 animals. *C1ql1$^{Cre}$*; *Sox10$^{DTA}$*: *n* = 3–4 animals. Unpaired *t* test with Welch's corrections. (B) PDGFRa immunolabeling of OPCs (magenta) reveals the decrease in OPC density in both regions of the corpus callosum at P7 while no effect is yet detected in the cortex in *C1ql1$^{Cre}$*; *Sox10$^{DTA}$* brains. Scale bars = 50 μm. Data are represented as mean ± SEM. *C1ql1$^{wt}$*; *Sox10$^{DTA}$*: *n* = 7 animals. *C1ql1$^{Cre}$*; *Sox10$^{DTA}$*: *n* = 3–4 animals. Unpaired *t* test with Welch's corrections. ns = not significant. The individual numerical values underlying this figure can be found in S1 Data.
(EPS)

**S5 Fig. Proliferation of *C1ql1$^{neg}$* OPCs is not increased upon ablation of *C1ql1*-expressing OPCs during the first postnatal week.** Proliferative cells were immunolabeled for Ki67 (magenta) in coronal sections of the cortex from P0 control or *C1ql1$^{Cre}$*; *Sox10$^{DTA}$* animals. Scale bars = 50 μm. Quantification of the density and the percentage of the proliferating oligolineage cells (Ki67$^{pos}$ GFP$^{pos}$, yellow arrowheads) shows no significant difference between both genotypes. Data are represented as mean ± SEM. *C1ql1$^{wt}$*; *Sox10$^{DTA}$*: *n* = 5–7 animals. *C1ql1$^{Cre}$*; *Sox10$^{DTA}$*: *n* = 4–6 animals. Unpaired *t* test with Welch's corrections. ns = not

significant. The individual numerical values underlying this figure can be found in S1 Data.
(EPS)

**S6 Fig. Ablation of *C1ql1*-expressing OPCs during postnatal development is with time compensated by GFP^neg OPCs.** (A) Coronal sections from adult brains of control (*C1ql1^wt; Sox10^DTA*) and *C1ql1^pos* OPC-ablated (*C1ql1^Cre; Sox10^DTA*) mice. Direct GFP fluorescence and Hoechst staining are shown. Scale bars = 1 mm. (B) OPCs were immunolabeled with antibodies against NG2 (white) and PDGFRa (magenta). NG2^pos PDGFRa^pos GFP^pos (yellow arrowhead) are detected in both genotypes. NG2^pos PDGFRa^pos GFP^neg (white arrowhead) corresponding to GFP^neg OPCs repopulate the forebrain in the adult after postnatal ablation of *C1ql1*-expressing OPCs. Scale bars = 100 μm.
(EPS)

**S7 Fig. *C1ql1* marks a differentiation stage distinct from the COP and NFOL stages.** (A) Network analysis shows the genes co-enriched with *C1ql1* in E13.5, P7, and juvenile adult oligolineage cells (reanalysis of raw data from Marques and colleagues [19]). (B) New analysis of the single-cell transcriptomic raw data from Marques and colleagues [19] shows the overlap of the expression of *C1ql1* (first left panel) with the OPC marker *Cspg4* (coding NG2), but not with the COP and NFOL markers *Gpr17*, *Bcas1*, and *Pcdh17it*. UMAP, uniform manifold approximation and projection.
(EPS)

**S8 Fig. Genetic ablation of *C1ql1*-expressing OPCs prevents myelination in the forebrain without affecting axogenesis.** (A) Neuronal fibers undergoing myelination were immunolabeled with an antibody against CNP (red), an early marker of myelination, in coronal sections from P16 brains (CC—center: central part of corpus callosum; CC—lateral: lateral part of corpus callosum). Quantification of the thickness of the myelinated bundles was performed in the corpus callosum, while in the cortex the global level of myelination was assessed by measuring the raw integrated density of labeling. CNP is almost absent in the lateral part of corpus callosum and cortex after ablation of *C1ql1*-expressing OPCs. Data are represented as mean ± SEM. *C1ql1^wt; Sox10^DTA*: $n = 5$ animals. *C1ql1^Cre; Sox10^DTA*: $n = 3–4$ animals. Unpaired $t$ test. ns = not significant. Scale bars = 100 μm. (B) Neuronal fibers were immunolabeled with an antibody against neurofilament (NF, green) in coronal brain sections of P16 mice. Quantification of the thickness of the fiber bundles in the corpus callosum and the intensity of NF in the cortex shows a small decrease in the thickness of the lateral part of the corpus callosum, probably explained by reduced myelination. Data are represented as mean ± SEM. *C1ql1^wt; Sox10^DTA*: $n = 3$ animals. *C1ql1^Cre; Sox10^DTA*: $n = 3$ animals. Unpaired $t$ test. ns = not significant. Scale bars = 100 μm. The individual numerical values underlying this figure can be found in S1 Data.
(EPS)

**S9 Fig. *Sox10*-GFP^neg OPCs do not rescue the myelination phenotype in adult mice.** (A) Low magnification of a coronal brain section of adult mouse immunostained for myelin basic protein (MBP, magenta) and 2′,3′-Cyclic nucleotide 3′-phosphodiesterase (CNP, red) illustrates the almost total absence of myelination in the dorsal forebrain upon genetic ablation of *C1ql1*-expressing OPCs (upper panel). Scale bars = 1,000 μm. Higher magnification of the coronal sections in the center and lateral parts of CC and cortex immunostained for MBP (magenta) and CNP (red) shows the dramatic decrease or absence of myelination in the lateral part of corpus callosum and the cortex of adult *C1ql1^Cre; Sox10^DTA*. Scale bars = 100 μm. (B) Staining for FluoroMyelin Red (green) and imaging with spectral confocal reflectance microscopy (SCoRe, white) in coronal sections shows lack of compact myelin in the lateral corpus

callosum and cortex of adult *C1ql1^{Cre}; Sox10^{DTA}* mice. Scale bars = 100 μm.
(EPS)

**S1 Data. Raw data (individual numerical values of all replicates, their means, and errors) underlying Figs 1C, 2C, 2D, 3A, 3C, 4B, S3C, S4A, S4B, S5, S8A and S8B.**
(XLSX)

## Acknowledgments

We would like to thank Lamia Bouslama-Oueghlani and Noémie Adès from ICM, Paris for their scientific suggestions; Philippe Mailly and Héloïse Monnet from CIRB Imaging facility (ORION) for their contribution in developing the home-made plugins to quantify the smFISH and immunofluorescence data; William D. Richardson from UCL, London for *Sox10^{DTA}* mice and scientific suggestion; the Venance team for sharing the *R26^{R-EYFP}* mouse line; and CIRB administration and animal facility personnel.

## Author Contributions

**Conceptualization:** Shayan Moghimyfiroozabad, Fekrije Selimi.

**Formal analysis:** Lea Bellenger, Fekrije Selimi.

**Funding acquisition:** Shayan Moghimyfiroozabad, Fekrije Selimi.

**Investigation:** Shayan Moghimyfiroozabad, Maela A. Paul, Fekrije Selimi.

**Methodology:** Lea Bellenger, Fekrije Selimi.

**Project administration:** Fekrije Selimi.

**Supervision:** Fekrije Selimi.

**Writing – original draft:** Shayan Moghimyfiroozabad, Fekrije Selimi.

**Writing – review & editing:** Shayan Moghimyfiroozabad, Maela A. Paul, Lea Bellenger, Fekrije Selimi.

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
