## [Editor Report · Decision Letter 0]

10 Aug 2023

Dear Fekrije, 

Thank you for submitting your manuscript entitled "A molecularly-defined non-redundant subpopulation of OPCs controls the generation of myelinating oligodendrocytes during postnatal development." for consideration as a Discovery Report by PLOS Biology.

Your manuscript has now been evaluated by the PLOS Biology editorial staff as well as by an academic editor with relevant expertise and I am writing to let you know that we would like to send your submission out for external peer review.

Once your full submission is complete, your paper will undergo a series of checks in preparation for peer review. After your manuscript has passed the checks it will be sent out for review. To provide the metadata for your submission, please Login to Editorial Manager (https://www.editorialmanager.com/pbiology) within two working days, i.e. by Aug 12 2023 11:59PM.

Kind regards,

Christian

Christian Schnell, PhD

Senior Editor

PLOS Biology

cschnell@plos.org

---

## [Decision Letter · Decision Letter 1]

4 Oct 2023

Dear Fekrije,

Thank you for your patience while your manuscript "A molecularly-defined non-redundant subpopulation of OPCs controls the generation of myelinating oligodendrocytes during postnatal development." was peer-reviewed at PLOS Biology. It has now been evaluated by the PLOS Biology editors, an Academic Editor with relevant expertise, and by several independent reviewers. 

In light of the reviews, which you will find at the end of this email, we would like to invite you to revise the work to thoroughly address the reviewers' reports.

As you will see below, the reviewers think that the study is very well executed and provides important insights, despite a lack of mechanistic insight. However, both reviewers raise a couple of points where further experimental data are necessary to support the claims.

Given the extent of revision needed, we cannot make a decision about publication until we have seen the revised manuscript and your response to the reviewers' comments. Your revised manuscript is likely to be sent for further evaluation by all or a subset of the reviewers.

**IMPORTANT - SUBMITTING YOUR REVISION**

*Re-submission Checklist*

* Please note that we allow a maximum of four figures for Discovery Reports.

*Published Peer Review*

*PLOS Data Policy*

*Blot and Gel Data Policy*

Sincerely,

Christian

Christian Schnell, PhD

Senior Editor

PLOS Biology

cschnell@plos.org

REVIEWS:

Reviewer's Responses to Questions

Reviewer #1: No

Reviewer #2: Yes: Tobias Merson

Reviewer #1: This manuscript by Moghimyfiroozabad and colleagues presents c1ql1 as a novel marker for a subset of oligodendrocyte precursor cells (OPCs) that are crucial for myelination. They show that c1ql1 is expressed by approximately 50% of OPCs and that this population does not segregate to specific OPC clusters in single cell RNA sequencing data, nor does it overlap with established subgroup markers of committed or early differentiating cells. Ablation of c1ql1-expressing OPCs using an oligodendrocyte lineage specific DTA model results in a substantial lack of myelination despite the presence of other OPCs. Therefore, the authors conclude that c1ql1 defines a non-redundant OPC population that is crucial for the generation of myelinating oligodendrocytes, as stated in the title.

Work over the past 5 years or so has shown that oligodendrocyte precursors are not a homogenous population. They can exist side by side in different states as they progress along their lineage, and they can form functionally different groups depending on the local microenvironment that they reside in. Despite these experimental findings, there are still large gaps in our understanding on the roles and the identity of OPCs, largely because of the lack of markers that allow to distinguish OPC populations apart from markers of differentiation.

The present manuscript provides a novel molecular marker which in itself will be of great interest and benefit to the glial community. Moreover, it is a striking result to show how the loss of some OPCs leads to a near-complete absence of myelin. Although there are of course a lot of questions remaining as to how and why these phenotypes occur, and what the function of c1ql1 in OPCs might be, I regard the novelty and interest of the present data as they are as very high.

I only have one major comment which I think should be addressed with additional data: 

1. The statement that the remaining OPCs cannot compensate the loss of the c1ql1-pos OPCs needs to be substantiated with additional analysis because this statement is not valid from a single timepoint analysis at P16. The authors come to this conclusion because they refer to earlier work from the Richardson lab which has shown that ablation of all forebrain OPCs is compensated by a similar timepoint through repopulation by midbrain-derived OPCs. However, one needs to bear in mind that the timing of the ablation is different. In the Richardson work, OPCs were eliminated from the very beginning (immediately after their specification in embryo and around birth), meaning that there was more time for compensation than in the experiments carried out here.

What the authors observe looks to me more like a timing effect with OPCs dying later, and limited by the circumstance that the animals do not survive much beyond P16. There are multiple ways to gain more insight into the capacity of compensation (or its lack of) and I cannot tell which one will be feasible for the authors. For example, how long does it take for OPCs to die from the moment they begin to express c1ql1 in the DTA ablation model. If death is not instantaneously, there would be even less time for OPC compensation. This would explain why there is no OPC reduction in the cortex but in the cc at P7 whereas there is some in the cc; presumably c1ql1 cells are present in the cc earlier than in the Ctx.

What should be feasible though is a more careful analysis of when c1ql1-expressing OPCs arrive in these different CNS regions, the timing of their disappearance, and analysis of the proliferative responses of the c1ql1-negative OPCs at different timepoints before P16. This would help understand the presented findings better. It would also very informative to learn about OPC numbers and potential compensation in animals that do survive longer that P16, even if n numbers will be low. 

Because if c1ql1-OPCs cannot get compensated, it would mean that they are entirely distinct from the remaining OPCs, which is highly unlikely given that these cells can be found across different OPC clusters. It is probably more likely that any OPC can, in principle, become c1ql1 positive, depending on yet unknown cues (this would also be a worthy discussion point(!)).

Minor points:

2. could the authors discuss why c1c1l /sox-DTA mice die so early, which is unlikely due to the lack of some OPCs and myelination reported here? Do they know if there are other cell types ablated outside the CNS? 

3. it would help the reader if the authors could provide rough numbers/fractions of effect sizes throughout the results section. For example, at the beginning of the results section. What is the proportion of c1ql1 pos cells which are NeuN-negative? This would help the reader to get a better idea of significance of these observation.

4. out of curiosity, a recent study of human white matter snRNA seq has also described two OPC subpopulations (Seeker et al., 2023; https://doi.org/10.1186/s40478-023-01568-z). Is it possible to tell how do the OPC_A and OPC_B populations seen in this other study compare to the OPC_1 and OPC_2 populations described here?

5. for better readability, it would be great if the authors could consider reworking their figures: harmonise font sizes used, use headings within the figure panels so that one can grasp quicker what is being assessed here.

Reviewer #2: This paper investigates the extent to which genetic ablation of oligodendroglia defined by the expression of Sox10 and C1ql1 perturbs homeostatic control of the OPCs during postnatal development, and their capacity to generate oligodendrocytes. They present evidence that C1ql1 is expressed in a subpopulation of OPCs that appear to be poised to differentiate into oligodendrocytes and that ablating these C1ql1+ oligodendroglia during postnatal development results in OPC depletion and a marked reduction in oligodendrocyte density and myelination. Whilst the paper demonstrates that C1ql1 expression is likely an essential pre-requisite for the generation of oligodendrocytes during postnatal myelination, due to the nature of the genetic manipulation, it does not provide any insight into how C1ql1 is acting in OPCs or whether C1ql1 expression is required for postnatal myelination. 

The authors demonstrate by crossing a C1ql1Cre knockin mouse line with the Cre-dependent R26Cas9-GFP line, that, in addition to neuronal populations, a subpopulation of NG2+ OPCs express GFP, indicative of Cre-mediated recombination. Additional evidence supporting the expression of C1ql1 by OPCs in wildtype mice is provided by duplex smFISH analysis which reveals that approximately 50-60% of NG2+ cells co-express C1ql1. The authors reanalyze published RNA-seq data that define the transcriptome of oligodendroglia in juvenile and adult mice to corroborate their expression data. Next, by crossing the C1ql1Cre line with the Cre-dependent Sox10DTA mouse line, the authors observed a ~75% and ~30% reduction in OPC density in the corpus callosum and cerebral cortex respectively, due to the induction of DTA expression in C1ql1-expressing OPCs. Intriguingly, whilst C1ql1 mRNA is present in ~70% of cortical OPCs at P7 and P15, a reduction in OPC density was not observed at P7, and was only reduced by 30% at P16. By contrast, OPC density in the corpus callosum is already diminished at P7 and the degree of this reduction more closely reflects the proportion of OPCs in this compartment that expresses C1ql1. The authors claim that this discrepancy reflects a selective failure of compensation for OPC loss in the corpus callosum. An alternate explanation is that C1ql1 expression in the corpus callosum precedes that in the cortex, and that recombination of the Sox10DTA allele occurs later in the cortex, meaning that DTA expression in OPCs in this region is delayed relative to the corpus callosum. To support their argument for differences in compensatory mechanisms, the authors should provide evidence that cortical OPCs in C1ql1Cre; Sox10DTA mice are indeed undergoing apoptosis at P7. The authors also suggest that since OPC migration into the dorsal forebrain is complete by P10, compensation for OPC loss does not occur beyond this time-point. This conclusion is inconsistent with their own data showing lack of compensation for OPC loss in the corpus callosum at P7 and evidence from adult OPC ablation experiments showing that partial OPC ablation in both gray and white matter can be compensated by residual non-ablated OPCs (Nat Neurosci. 2013 Jun;16(6):668-76, Cell Reports Meth. 2023 Feb 28;3(2):100414). 

Next, the authors perform Gene Ontology analysis of a previously published RNA-seq dataset which provides evidence that C1ql1 expression by OPCs may reflect a specific transition state of OPCs poised to differentiate into oligodendrocytes but declines at the point at which newly formed oligodendrocytes are specified. Indeed, since C1ql1Cre; Sox10DTA mice exhibit an even higher degree of CC1+ cell loss than might be expected from the proportion of OPCs that express C1ql1, this raises the possibility that C1ql1 is expressed at a necessary transition step during the differentiation of OPCs into oligodendrocytes. If C1ql1 is indeed a late marker of OPCs that are poised to differentiate, one might anticipate that this expression coincides with a flux in the expression of genes associated with cell division. It would therefore be valuable for the authors to explore whether transcripts for mitosis-associated proteins differ between C1ql1-negative and C1ql1-positive subpopulations of OPCs. This is also of particular interest given the previous data that links C1QL1 to cell proliferation and tumorigenicity. 

Whilst the examination of CC1+ cell density in the cortex and corpus callosum provides compelling evidence that oligodendrocyte density is markedly reduced in C1ql1Cre; Sox10DTA mice, the examination of the effects upon myelin formation is less conclusive. Their analysis is complicated in part by focusing on immunohistochemical analysis of MBP expression, a myelin protein that is expressed at a relatively early stage of oligodendrocyte differentiation. Their data show significant residual MBP expression in the corpus callosum which leaves it unclear how myelination per se is

---

## [Decision Letter · Decision Letter 2]

6 Mar 2024

Dear Fekrije,

Thank you for your patience while we considered your revised manuscript "A molecularly-defined subpopulation of OPCs controls the generation of myelinating oligodendrocytes during postnatal development." for consideration as a Discovery Report at PLOS Biology. Your revised study has now been evaluated by the PLOS Biology editors, the Academic Editor and the original reviewers.

In light of the reviews, which you will find at the end of this email, we are pleased to offer you the opportunity to address the remaining points from the reviewers in a revision that we anticipate should not take you very long. We will then assess your revised manuscript and your response to the reviewers' comments with our Academic Editor aiming to avoid further rounds of peer-review, although might need to consult with the reviewers, depending on the nature of the revisions.

**IMPORTANT - SUBMITTING YOUR REVISION**

*Resubmission Checklist*

*Published Peer Review*

*PLOS Data Policy*

*Blot and Gel Data Policy*

Sincerely,

Christian

Christian Schnell, PhD 

Senior Editor

PLOS Biology

cschnell@plos.org

REVIEWS:

Reviewer #1: The authors have fully addressed all my points from the previous review. No further comments.

Congratulations on this interesting work.

Reviewer #2 (Tobias D. Merson): The authors have conducted additional experiments and analyses that have addressed most of the concerns that I raised in my previous review and the manuscript is much improved. However, I have one major concern with the authors' interpretation of some of their data and the terminologies they use to describe them. The authors posit (lines 7-13, p.8) that the emergence of Sox10-GFPneg PDGFRapos NG2pos cells in the adult C1ql1Cre; Sox10DTA mouse brain suggests their origin from neural progenitor cells within the ventricular-subventricular zone (V-SVZ), rather than from Sox10 lineage-derived OPCs. They refer to these Sox10-GFPneg PDGFRapos NG2pos as "non-Sox10 lineage derived OPCs". However, this conclusion presents several challenges. To bolster their claim, the authors reference a study by Xing et al. (Cell Rep. Methods 3, 100414, 2023) which details the pharmacogenetic ablation of OPCs using Pdgfra-CreERT2; Sox10DTA mice coupled with intracisternal infusion of the anti-mitotic AraC. In this study, Xing and team illustrate that PDGFRA+ cells regenerated in the cerebrum after OPC ablation originate from neural precursor cells born in the V-SVZ. However, these NPC-derived OPCs were demonstrated to express GFP, indicating that the transgenic Sox10 promoter is active in OPCs that derive from the V-SVZ. Moreover, in contrast with the authors' claim that all oligolineage cells in Sox10DTA mice express GFP (lines 3-4, 22-23, p.5), the original characterization of Sox10DTA mice (Kessaris et al., 2016) highlights that some 2% of OL lineage cells do not co-express GFP. Additionally, Xing et al. (2023) also report that some OPCs in adult Sox10DTA mice do not express GFP, despite expression of endogenous SOX10 protein. In fact, contrary to their claim that all OL lineage cells express GFP, the authors highlight the presence of GFPneg CC1pos cells in the center of the corpus callosum of C1ql1Cre; Sox10DTA (line 14, p. 9). These observations demonstrate that the transgenic Sox10 promoter in Sox10DTA mice, which drives GFP expression in non-recombined cells and DTA following Cre-mediated recombination, isn't universally active in all OL lineage cells. Moreover, the authors' use of the term "non-Sox10 lineage derived OPCs" is problematic as they only show that these cells do not express GFP; they have not shown that they fail to express SOX10 protein, which would be necessary to support the use of this terminology. A more accurate term would be "GFPneg OPCs". The most plausible interpretation of the authors' findings in adult C1ql1Cre; Sox10DTA mice is the selective expansion of OPC clones where the transgenic Sox10 promoter remains inactive. Consequently, these OPCs fail to express either GFP or DTA. Considering that NPC-derived OPCs display activity of the transgenic Sox10 promoter, it follows that any NPC-derived OPCs that are generated after OPC depletion in C1ql1Cre; Sox10DTA mice, would also be expected to undergo DTA-mediated apoptosis. The authors' observation that surviving adult C1ql1Cre; Sox10DTA mice exhibit marked deficiency in OLs and myelin, despite an expansion in the population of GFPneg OPCs, raises the intriguing possibility that these GFPneg OPCs could be defective in their differentiative capacity in this context. The results and discussion sections of the manuscript need to be revised extensively to address these issues regarding the interpretation of their data.

---

## [Decision Letter · Decision Letter 3]

17 Apr 2024

Dear Fekrije,

Thank you for your patience while we considered your revised manuscript "A molecularly-defined subpopulation of OPCs controls the generation of myelinating oligodendrocytes during postnatal development." for publication as a Discovery Report at PLOS Biology. This revised version of your manuscript has been evaluated by the PLOS Biology editors, the Academic Editor and the original reviewers.

Based on the reviews, we are likely to accept this manuscript for publication, provided you satisfactorily address the following data and other policy-related requests.

* We had discussed earlier that we will leave the decision about the appropriate format for a later stage. I have now discussed this point with my colleagues and we recommend that our Short Reports format is the most suitable format for your paper, given the increased depth during the review process and other details (such as the number of figures). Please do not hesitate to let me know if you have any questions/concerns about this. 

* We would like to suggest a different title to improve readability: "A molecularly-defined subpopulation of oligodendrocyte precursor cells controls the generation of myelinating oligodendrocytes during postnatal development"

* Please add the links to the funding agencies in the Financial Disclosure statement in the manuscript details.

DATA POLICY:

Regardless of the method selected, please ensure that you provide the individual numerical values that underlie the summary data displayed in the following figure panels as they are essential for readers to assess your analysis and to reproduce it: 1D, 2C, 2D, 3A, 3C, 4B, S3C, S4A, S4B, S5, and S8.

CODE POLICY

Per journal policy, if you have generated any custom code during the curse of this investigation, please make it available without restrictions upon publication. Please ensure that the code is sufficiently well documented and reusable, and that your Data Statement in the Editorial Manager submission system accurately describes where your code can be found. 

We expect to receive your revised manuscript within two weeks. 

*Published Peer Review History*

*Press*

Sincerely,

Christian

Christian Schnell, PhD

Senior Editor

cschnell@plos.org

PLOS Biology

Reviewer remarks:

Reviewer #2 (Tobias D. Merson): The authors have addressed the concerns detailed in my previous review and have revised the manuscript accordingly.

---

## [Editor Report · Decision Letter 4]

2 May 2024

Dear Fekrije,

Thank you for the submission of your revised Short Reports "A molecularly-defined subpopulation of oligodendrocyte precursor cells controls the generation of myelinating oligodendrocytes during postnatal development." for publication in PLOS Biology. On behalf of my colleagues and the Academic Editor, Mikael Simons, I am pleased to say that we can in principle accept your manuscript for publication, provided you address any remaining formatting and reporting issues. These will be detailed in an email you should receive within 2-3 business days from our colleagues in the journal operations team; no action is required from you until then. Please note that we will not be able to formally accept your manuscript and schedule it for publication until you have completed any requested changes.

PRESS

Sincerely, 

Christian Schnell

Christian Schnell, PhD

Senior Editor

PLOS Biology

cschnell@plos.org